# Autoregressive Policy Optimization for Constrained Allocation Tasks

**David Winkel**[*]   **Niklas Strauß**[*]
**Maximilian Bernhard**   **Zongyue Li**   **Thomas Seidl**   **Matthias Schubert**
Munich Center for Machine Learning, LMU Munich
{winkel,strauss,bernhard,li,seidl,schubert}@dbs.ifi.lmu.de

## Abstract

Allocation tasks represent a class of problems where a limited amount of resources must be allocated to a set of entities at each time step. Prominent examples of this task include portfolio optimization or distributing computational workloads across servers. Allocation tasks are typically bound by linear constraints describing practical requirements that have to be strictly fulfilled at all times. In portfolio optimization, for example, investors may be obligated to allocate less than 30% of the funds into a certain industrial sector in any investment period. Such constraints restrict the action space of allowed allocations in intricate ways, which makes learning a policy that avoids constraint violations difficult. In this paper, we propose a new method for constrained allocation tasks based on an autoregressive process to sequentially sample allocations for each entity. In addition, we introduce a novel de-biasing mechanism to counter the initial bias caused by sequential sampling. We demonstrate the superior performance of our approach compared to a variety of Constrained Reinforcement Learning (CRL) methods on three distinct constrained allocation tasks: portfolio optimization, computational workload distribution, and a synthetic allocation benchmark. Our code is available at: `https://github.com/niklasdbs/paspo`.

## 1 Introduction

Continuous allocation tasks are a class of problems where an agent needs to distribute a limited amount of resources over a set of entities at each time step. Many complex real-world problems are formulated as allocation tasks, and state-of-the-art solutions rely on using Reinforcement Learning (RL) to learn effective policies [6, 26, 3, 20, 27]. Notable examples include portfolio allocation tasks, where portfolio managers must allocate the available financial resources among various assets [27], or allocation tasks of computational workloads to a set of compute instances in data centers [3]. In many cases, allocation tasks come with allocation constraints [6, 20, 27, 26], such as investing at most 30 % of the portfolio into a specific subset of the assets or to restrict the maximum workload to certain servers in a data center. Formally, allocation constraints are expressed as linear constraints and form a system of linear inequalities, geometrically describing a convex polytope. Each point in this polytope describes a possible allocation and each dimension corresponds to one of the entities. Allocation tasks often require hard constraints, i.e., constraints that are explicitly given and must be satisfied at any point in time. However, most of the existing CRL literature focuses on soft constraints that are not explicitly given [2, 29, 31, 14, 25]. These approaches typically cannot guarantee constraint satisfaction and tend to have many constraint violations during training. The majority of these methods approximate the cumulative costs of constraint violations and optimize the cumulative reward while trying to adhere to the maximum cumulative costs. While less explored, there exist

---

[*]Both authors contributed equally.

38th Conference on Neural Information Processing Systems (NeurIPS 2024).

several techniques that ensure the satisfaction of hard constraints [18, 6, 11, 10, 20]. These approaches might generate actions that do not satisfy the constraints but utilize a correction mechanism to map the actions back into the valid action space. In addition, most of these approaches are restricted to off-policy algorithms [11, 20]. In another line of research, solutions tailored for constrained allocation tasks have been proposed [27, 28]. However, these solutions are severely limited since they can only handle a specific subset of linear constraints and cannot handle more than two.

In this paper, we propose Polytope Action Space Policy Optimization (PASPO), a novel RL-based method that, firstly, decomposes the action space into several dependent sub-problems and, secondly, autoregressively computes the allocations step-by-step for each entity individually. In contrast to previous methods for hard constraints, we directly generate an action within the action space. This makes the correction of invalid actions unnecessary and, thus, avoids potential sampling bias introduced by the correction. Our new decomposition approach is implemented in a neural network-based policy function, which can be employed in on-policy and off-policy RL algorithms. We show that initialization bias can prevent proper exploration in early training which leads to premature convergence. Thus, we propose a de-biasing mechanism to stabilize exploration in early training stages.

We evaluate our approach against various baselines on three distinct allocation tasks: portfolio optimization, distributing computational workload in data centers, and a synthetic benchmark. These experiments demonstrate that our approach can outperform existing methods consistently and show the importance of the proposed de-biasing mechanism.

To summarize, the main contributions of our paper are:

- A new autoregressive stochastic policy function applicable to arbitrary convex polytope action spaces of constrained allocation tasks.
- A new de-biasing mechanism to prevent premature convergence to a sub-optimal policy.
- An empirical evaluation that optimizes our new policy function using PPO [22] and demonstrates improved results compared to state-of-the-art CRL methods.

The remainder of the paper is structured as follows: In Section 2, we provide an overview of the related work in CRL, constrained allocation tasks, and autoregressive policy functions. Afterward, we formalize constrained allocation tasks in Section 3 and present our novel approach in Section 4. Section 5 describes the results of our experimental evaluation, Section 6 briefly discusses limitations and future work before Section 7 concludes the paper.

## 2 Related Work

Resource allocation tasks are a widely researched area with numerous applications spanning logistics, power distribution, computational load balancing, security screening, and finance [6, 26, 3, 20, 27]. We identify three key research directions that are particularly important when discussing resource allocation tasks.

**Safe Reinforcement Learning** The majority of work in CRL addresses soft constraints, a setting often referred to as Safe RL. We will provide a brief overview of the most important methods in this field. For a more comprehensive examination of Safe RL, we direct readers to the survey papers by [15, 12]. A common technique in Safe RL is the use of Lagrangian relaxation [4, 15]. Several works employ primal-dual optimization to leverage the Lagrangian duality, including [9, 23, 13]. Another frequently used approach involves different penalty terms [25, 14, 31]. The authors of IPO [14] propose to use logarithmic barrier functions. CPO [2] extends TRPO [21] to ensure near-constraint satisfaction with each update. Additionally, two-step approaches such as FOCOPS [32] and CUP [30] are popular in the field. However, unlike our method, these approaches do not guarantee strict constraint satisfaction, particularly during training.

**Hard Constraints** Although less studied than Safe RL, several works address hard instantaneous constraints on actions to ensure full constraint satisfaction at any time step. Most of these approaches employ mechanisms to correct infeasible actions, i.e., those that violate constraints, into feasible actions [18, 6, 20, 11]. In contrast, our method always generates feasible actions without the need for correction. OptLayer [18] is one of the most prominent examples in this field, which employs OptNet [5] to map infeasible actions to the nearest feasible action. Similarly, [20] propose a more

efficient projection on the polytope action space than OptLayer. The authors of [6] focus on resource allocation with hierarchical allocation constraints by proposing a faster approximate version of OptLayer. In [11], the authors propose an off-policy algorithm based on the generalized reduced gradient method [1] to handle non-linear hard constraints by projecting infeasible actions. In contrast, our method is not limited to off-policy algorithms.

In [27, 28], the action space is decomposed into independent subspaces. However, these approaches can only handle up to two allocation constraints. Furthermore, they are only applicable to binary allocation constraints. In contrast, our approach can handle an arbitrary number of constraints as well as any type of linear allocation constraints.

**Action Space Decomposition/Factorization** The decomposition or factorization of multi-dimensional action spaces has been examined in several works [24, 16, 19]. A notable example is [24], in which the authors discretize a continuous action space into several independent action branches, each parameterized by individual network branches. In [16], a variant of DQN [17] that discretizes a continuous action space into multiple discrete dimensions is proposed. These dimensions are sequentially parameterized, conditional on the previous sub-actions. Similarly, [19] propose an autoregressive factorization of an unconstrained action space into dependent sub-problems. Unlike our approach, these methods focus either on decomposing continuous action spaces into discrete action spaces or decomposing unconstrained action spaces. However, the decomposition of arbitrary convex polytope action spaces into tractable sub-action spaces remains a non-trivial challenge that our approach addresses.

## 3 Problem Description

An allocation task can be described as a finite-horizon Markov decision process (MDP) $(S, A, T, R, \gamma)$, where $S$ represents the state space, $A$ the action space, $T : S \times A \times S \rightarrow [0, 1]$ the state transition function, $R$ the reward function, and $\gamma \in [0, 1]$ a discount factor. The goal of this task is to find a policy $\pi$ maximizing the expected cumulative reward $J_R^\pi = \mathbb{E}_\pi \left[ \sum_{t=1}^n \gamma^t R(s_t, \pi(s_t), s_{t+1}) \right]$.

The action $a$ is an allocation $a = \{a_1, \ldots, a_n\} \in A$ over a set of $n$ entities $E = \{e_1, \ldots, e_n\}$ at each time step. Each element $a_i$ of the action vector $a$ represents the proportion allocated to entity $e_i$. Furthermore, allocation tasks require a complete allocation, i. e., $\sum_{i=1}^n a_i = 1$ and allocations cannot be negative ($a_i \geq 0$). Thus, the action space of unconstrained allocation tasks forms an $n$-dimensional standard simplex. A visualization of an unconstrained allocation action space is provided in Figure 1a.

Allocation tasks frequently include constraints, such as allocating at most 30% to a subset of the entities. An example of a constrained action space is visualized in Figure 1b. Formally, an allocation constraint can be expressed as a linear inequality $\sum_{i=1}^n c_i a_i \leq b$, where $c_i$ denotes the weighting of the allocation variable $a_i$ of entity $e_i$ and $b \in \mathbb{R}$ denotes the corresponding constraint limit. For the sake of readability and simplicity, we only define $\leq$ constraints since $a \geq b$ can be transformed into $-a \leq -b$ and $a = b$ can be rewritten as $a \leq b$ and $-a \leq -b$.

The action space $A$ of constrained allocation tasks can be easily expressed by a set of linear inequalities, defining a polytope $A = \{a \in [0, 1]^n | Ca \leq b\}$, where

$$C \in \mathbb{R}^{m \times n} = \begin{bmatrix} c_{11} & \ldots & c_{1n} \\ \vdots & \ddots & \vdots \\ c_{m1} & \ldots & c_{mn} \end{bmatrix} \tag{1}$$

is a matrix of coefficients for the $m$ constraints, including those linked to the simplex constraints $\sum_{i=1}^n a_i = 1$ and $a_i \geq 0 \; \forall i \in \{1, \ldots n\}$, as well as all coefficients for additional allocation constraints. Let $a \in [0, 1]^n$ represent an allocation vector and $b \in \mathbb{R}^m$ is the vector of constraint limits.

Alternatively, constrained allocation tasks can be defined using the framework of constrained Markov decision processes (CMDPs). A CMDP extends the standard MDP by a number of cost functions to incorporate the constraints. The goal is to maximize the expected cumulative reward while satisfying $m$ constraints on the expected cumulative costs. The expected cumulative costs for the $k$-th cost function $CF_k$ are defined as $J_{CF_k}^\pi = \mathbb{E}[\sum_{t=0}^T \gamma^t CF_k(s_t, a_t)]$. The $m$ constraints to be satisfied in the CMDP are then stated as $J_{CF_k}^\pi \leq d_k$, where $d_k$ denotes the cost limit with

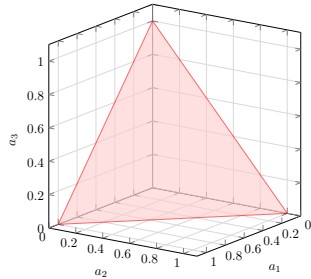
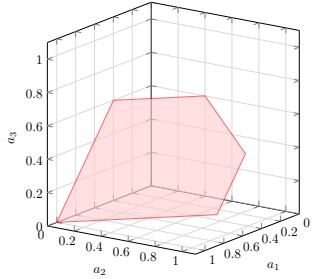

(a) Unconstrained standard simplex          (b) Constrained simplex with $a_3 \leq 0.6$ and $a_2 \leq 0.7$

Figure 1: **Examples of 3-dimensional allocation action spaces** (a) unconstrained and (b) constrained (valid solutions as red area).

$k \in \{1, \dots, m\}$. To formulate constrained allocation tasks using CMDPs, the cost functions can be defined as $CF_k(s,a) = \max\{0, (Ca)_k - b_k\}$ to measure any allocation constraint violation as a cost. In addition, strict adherence to all allocation constraints at any point in time is required, i.e., $d_k = 0$. By formulating constraint allocation tasks using CMDPs, it becomes possible to use existing methods from Safe RL for soft constraints. However, these methods cannot guarantee constraint satisfaction at all times [15]. Let us note that our method does not use cost functions, instead it samples actions directly from the constrained action space.

## 4 Polytope Action Space Policy Optimization (PASPO)

Our approach PASPO autoregressively computes the allocation to every single entity in an iterative process until all allocations are fixed. We will later show that this step-wise decomposition allows for a tractable parametrization of the action space.

### 4.1 Autoregressive Polytope Decomposition

PASPO starts by determining the feasible interval $[a_1^{min}, a_1^{max}]$ for allocations into the first entity $e_1$. Then, we sample the first allocation $a_1$ from this interval. The details of the sampling process will be further discussed in Section 4.2. Fixing an allocation impacts the shape of the remaining action space. Thus, we have to compute the shape of the polytope $A^{(2)}$ described by $C^{(2)}$ and $b^{(2)}$ before we can sample the next allocation $a_2$.

Each iteration $i$ starts with determining the interval $[a_i^{min}, a_i^{max}]$ of all feasible values for $a_i$. Geometrically, this interval is bounded by the minimum and the maximum value of the remaining polytope $A^{(i)}$ in the $i$-th dimension associated with the allocation $a_i$. To determine $a_i^{min}$, we solve the following linear program:

$$\text{minimize} \quad a_i$$
$$\text{s.t.} \quad C^{(i)} a^{(i)} \leq b^{(i)}$$

where $C^{(i)}$ are the constraint coefficients for the entities $e_i, \dots, e_n$, $b^{(i)}$ are the adjusted constraint limits, and $a^{(i)}$ describes the unfixed allocations. We determine $a_i^{max}$ by solving the respective maximization problem. For the first iteration $i = 1$, we define $C^{(1)} = C$, $b^{(1)} = b$ and $a^{(1)} = a$. After sampling an allocation $a_i$ from the interval $[a_i^{min}, a_i^{max}]$. The resulting polytope $A^{(i+1)}$ for the next iteration $i + 1$ is described by the following inequality system:

$$\underbrace{\begin{bmatrix} c_{1,i+1} & \cdots & c_{1,n} \\ \vdots & \ddots & \vdots \\ c_{m,i+1} & \cdots & c_{m,n} \end{bmatrix}}_{C^{(i+1)}} \underbrace{\begin{bmatrix} a_{i+1} \\ \vdots \\ a_n \end{bmatrix}}_{a^{(i+1)}} \leq \underbrace{\begin{bmatrix} b_1^{(i)} \\ \vdots \\ b_m^{(i)} \end{bmatrix} - a_i \begin{bmatrix} c_{1,i} \\ \vdots \\ c_{m,i} \end{bmatrix}}_{b^{(i+1)}} \tag{2}$$

To define the new coefficient matrix $C^{(i+1)}$ (red), we remove the first column of the coefficient matrix of the previous iteration $C^{(i)}$. To calculate the new vector $b^{(i+1)}$ of constraint limits, we subtract

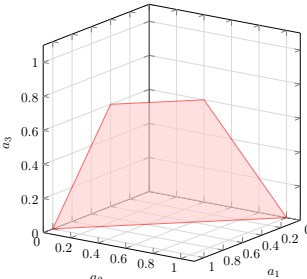
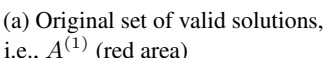
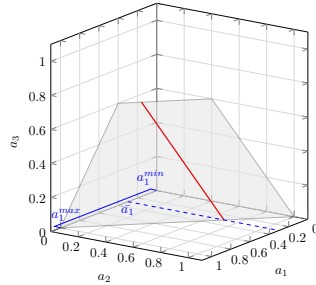
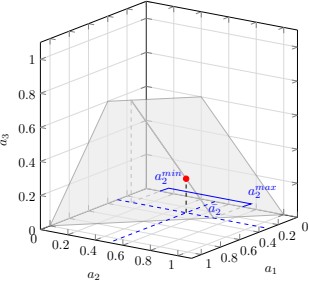

(a) Original set of valid solutions, i.e., $A^{(1)}$ (red area)

(b) Remaining valid solutions in $A^{(2)}$ (red line) after $a_1 = 0.3$ (dashed blue line) was fixed, i.e., sampled

(c) Only a single valid solution remains in $A^{(3)}$ (red dot) after $a_1 = 0.3$ and $a_2 = 0.5$ (dashed blue lines) are fixed, i.e., sampled

Figure 2: **Example of sampling process** of an action $a = (a_1, a_2, a_3)$ in a 3-dimensional constrained allocation task.

the removed column (blue) scaled by the fixed allocation $a_i$ from the previous constraint limits $b^{(i)}$ (yellow). We iterate over all entities until we determine $a_{n-1}$. Allocation $a_n$ is already determined as soon as the allocations $a_1, \ldots, a_{n-1}$ are fixed because of the simplex constraint $\sum_{i=1}^{n} a_i = 1$. Sampling an allocation using this approach always guarantees constraint satisfaction and it is possible to sample any action in the constrained action space. A formal proof of these guarantees can be found in Appendix D.

Figure 2 displays a visualization of the process for a 3-dimensional case. The set of valid solutions before any allocations have been fixed is shown in Figure 2a. Figure 2b depicts the first iteration after $a_1 = 0.3$ has been determined, and the resulting new polytope $A^{(2)}$, i.e., a set of valid solutions, shrinks to the red line is shown. Figure 2c shows the second iteration after also $a_2 = 0.5$ has been determined. It can be seen that the new polytope $A^{(3)}$ contains only a single valid solution represented as a red dot, making a third iteration unnecessary since the only remaining solution is to allocate $a_3 = 0.2$, resulting in a final allocation of $a = (0.3, 0.5, 0.2)$.

### 4.2 Parameterizable Policy Process

Our goal is to define a learnable stochastic policy function over the action space. For unconstrained allocation tasks, a Dirichlet distribution can be used to parameterize the action space [26, 28]. Unfortunately, to the best of our knowledge, there is no known parameterizable, closed-form distribution function over arbitrary convex polytopes as in our setting. In fact, even uniform sampling over a convex polytope is an active research problem [8].

We sequentially constructed an action $a$ from the polytope action space $A$ in the previous section. Now, we describe how to utilize this process to define a parameterizable policy function over the action space $A$. We model the distribution for allocating each individual entity using a beta distribution that is normalized to the range $[a_i^{min}, a_i^{max}]$. This distribution is also known as the four-parameter beta distribution [7]. Its probability density function is defined as:

$$p(x; \alpha, \beta, a_i^{min}, a_i^{max}) = \frac{(x - a_i^{min})^{\alpha-1}(a_i^{max} - x)^{\beta-1}}{(a_i^{max} - a_i^{min})^{\alpha+\beta-1}B(\alpha, \beta)},$$

where $B(\alpha, \beta)$ is the beta function. It is important to note that any other parameterizable distributions with bounded support in the range $[a_i^{min}, a_i^{max}]$ can be used, such as a squashed Gaussian distribution. However, our preliminary experiments indicated that the beta distribution performs particularly well.

To optimize the policy $\pi_\theta(s)$ over the complete allocations, we follow the approach of [19] for training an autoregressively dependent series of sub-policies. A fixed but arbitrary order of entities is used for sampling the allocations $a_i$. The sub-policy $\pi_\theta^i(a_i|a_1, \ldots, a_{i-1})$ is conditional on the previous allocations $a_1, \ldots, a_{i-1}$. Using this autoregressive dependence structure, the policy is defined as: $\pi_\theta(a|s) = \pi_\theta^1(a_1|s) \cdot \pi_\theta^2(a_2)|s, a_1) \ldots \pi_\theta^n(a_n|s, a_1, \ldots, a_{n-1})$. This policy can be jointly optimized. We parameterize each sub-policy using a neural network that receives an embedding of the state and the previously selected actions as input.

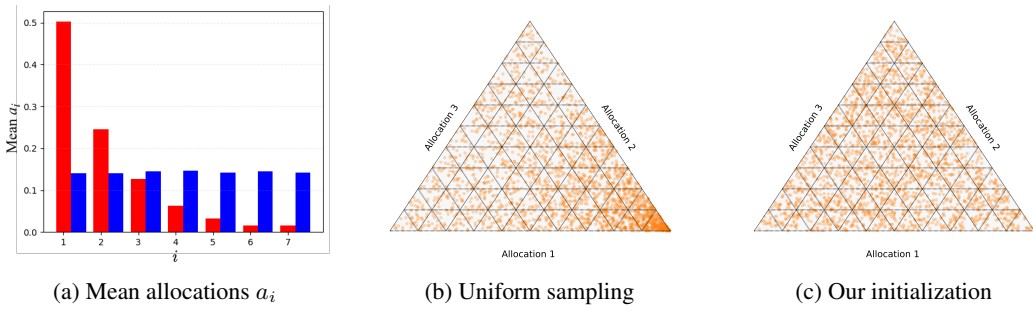

| (a) Mean allocations $a_i$ | (b) Uniform sampling | (c) Our initialization |
|---|---|---|

Figure 3: **The impact of initialization in an unconstrained simplex.** (a) Mean allocations $a_i$ to each entity in a seven entity setup when sampling each individual allocation using the uniform distribution (red) vs. our initialization (blue). (b,c) Distribution of 2500 allocations in a three entity setup when sampling each individual allocation uniformly (b) or using beta distributions with parameters set according to our initialization (c).

---

**Algorithm 1** Maximum likelihood estimation of parameter de-biasing terms
***

**Input**: Polytope $A = \{a \in \mathbb{R}_{0,+}^n | Ca \leq b\}$, number of samples $k$

**Output**: Beta distribution shape parameters $\hat{\alpha}_i, \hat{\beta}_i$ as de-biasing terms

1: Let $A_j^{(1)} = A$ for $j = \{1, \ldots, k\}$
2: Sample $k$ allocations $\{a_{(j)}\}_{j=1\ldots k}$ uniformly within $A$ via rejection sampling
3: **for** each dimension $i$ in $\{1, \ldots, n\}$ **do**
4:     **for** each sample $j$ in $\{1, \ldots, k\}$ **do**
5:         Calculate interval $[a_{(j)i}^{min}, a_{(j)i}^{max}]$ using LP based on $A_j^{(i)}$
6:         Normalize sampled allocation to support [0,1] of beta distribution: $a_{(j)i}^{norm} = \frac{a_{(j)i} - a_{(j)i}^{min}}{a_{(j)i}^{max} - a_{(j)i}^{min}}$
7:         Compute polytope $A_j^{(i+1)}$ from $A_j^{(i)}$ using sampled allocation $a_{(j)i}$ (see Eq. 2)
8:     **end for**
9:     ML estimation of beta distribution parameters $\hat{\alpha}_i, \hat{\beta}_i$ using $\{a_{(j)i}^{norm}\}_{j=1\ldots k}$
10: **end for**

---

An entropy term is often used to encourage exploration. However, our policy does not have a closed-form solution for entropy. Therefore, we follow [19] to empirically estimate the entropy:

$$H_{\text{emp}}(\pi_\theta(\cdot|s)) = \mathbb{E}_{a \sim \pi_\theta(\cdot|s)} \left[ \sum_{i=1}^n H\left(\pi_\theta^i\left(\cdot|s, a_1, \ldots, a_{i-1}\right)\right) \right] \quad (3)$$

Here, $H$ denotes the entropy of the beta distribution. We compute the expectation within each training batch to estimate the entropy of the complete policy function over the entropies of single actions. Let us note that when using an off-policy algorithm, the actions must be resampled using the current policy. As the current policy might have a significantly different parametrization than the sampling policy, we have to generate actions based on the current policy to estimate the entropy properly.

### 4.3 Policy Network Architecture

We create an embedding of the state using an MLP, denoted as $f_\theta(s) = MLP(s)$. We parameterize the probability distribution over allocations for each entity using an MLP $\pi_{\theta_i}^i(s) = MLP(f_\theta(s), a_1, \ldots, a_{i-1})$, which receives the latent encoding of the state and the previously sampled allocations $a_1, \ldots, a_{i-1}$ as input. Note that each of the MLPs $\pi_{\theta_i}^i$ has its own parameters. For further details, we refer to the Appendix.

### 4.4 De-biasing Mechanism

A drawback of generating actions by an autoregressive process is that a random initialization of the beta distributions leads to a sampling bias towards the entities selected earlier in the process. The

effect is caused by the autoregressive dependency structure of our process. To sample an allocation of 80% for entity $a_i$ the cumulated fixed allocations for earlier entities $\sum_{j=1}^{i-1} a_j$ must be at most 20%. However, this is rather unlikely if we initialize the distribution for all entities in a similar way. This effect can be observed in the red bars of Figure 3a. The red bars correspond to the average allocation for each dimension when uniformly drawing from an unconstrained seven-dimensional simplex with our autoregressive process for each entity $e_i$. As expected, the mean for the first dimension is $0.5$, which is the mean of a uniform distribution over the interval $[0, 1]$. Correspondingly, the mean is decreased by half for any successive further entity until entity 6, which has the same mean as entity 7 due to the simplex constraint. Even though the bias is more complex for constrained allocation spaces, a similar effect can be expected.

For policy gradient methods, such a bias in the initialization of the policy function can lead to convergence to poorly performing policies or long training times. As the initial policy is crucial for ensuring sufficient exploration of the state-action space, a biased initial distribution leads to underexplored regions in the state-action space. Consequently, well-performing actions might not be discovered. To counter this effect, we propose a de-biasing mechanism that adjusts the initial parameters of beta distributions to estimate a uniform sampling over the joint action space. During learning, the amount of required exploration decreases, and the parameters of the policy function are optimized to increase the cumulative rewards. Thus, the impact of our de-biasing mechanism should diminish over time. We achieve this effect by adding a de-biasing term to the linear layers' initial bias terms, predicting $\alpha_i$ and $\beta_i$ for entity $i$. As the default initialization of the bias terms has zero means, the first iterations use $\alpha$ and $\beta$ values close to the de-biasing terms.

To determine suitable initial values for each iteration step, we proceed as described in Algorithm 1. We start by uniformly sampling $n$ data points from the complete action space $A$. We do this by rejection sampling, i.e., we sample over the standard simplex and reject the samples outside the action polytope $A$. To determine the parameters corresponding to the acquired uniform sample, we project any allocation for each entity to a standard interval between $[0, 1]$. However, for this step, we have to determine the interval $[a_i^{min}, a_i^{max}]$ for each entity following the above process. We determine the relative position in this interval, corresponding to the position in the named standard interval. After collecting relative values for each sample and entity, we employ the standard maximum likelihood estimator to generate an empirical estimate of the $\alpha_i$ and $\beta_i$ for each entity $e_i$. The blue bars in Figure 3a correspond to the results on the unconstrained seven-dimensional unconstrained simplex. Figure 3b shows autoregressive sampling based on uniform distribution, whereas Figure 3c displays the result of our initialization for a three-dimensional example. It can be seen that the result of our initialization of the autoregressive process closely resembles a uniform distribution over the complete action space.

# 5 Experiments

In this section, we provide an extensive experimental evaluation of our approach in various scenarios demonstrating its ability to handle various allocation tasks and constraints. We use two real-world tasks: Portfolio optimization [27] and compute load distribution [3]. Additionally, we create a synthetic benchmark with a reward surface generated by a randomly initialized MLP. Each of these tasks comes with a different set of allocation constraints. We will briefly describe each setting in the following and refer the reader to the Appendix for more details.

**Portfolio Optimization** Portfolio optimization is a prominent constrained allocation task. In this task the agent has to allocate its wealth over 13 assets at each time step. We use the environment of [27]. Each investment period contains 12 months and the investor needs to reallocate the portfolio each month. This environment is highly stochastic since each trajectory is sampled from a hidden Markov model fitted on real-world NASDAQ-100 data. After every 5120 environment steps, we run eight parallel evaluations on 200 fixed trajectories. Constraints in this setting define minimum and maximum allocation to groups of assets. Additionally, we add constraints where the constraint coefficients in $C$ correspond to portfolio measures like a minimum dividend yield or a maximum on the CO2 intensity.

**Compute Load Distribution** The environment is based on the paper of [3] and simulates a data center in which computational jobs need to be split into sub-jobs to enable parallel processing across nine servers. Here, we use five constraints that are randomly sampled as follows: First, we sample

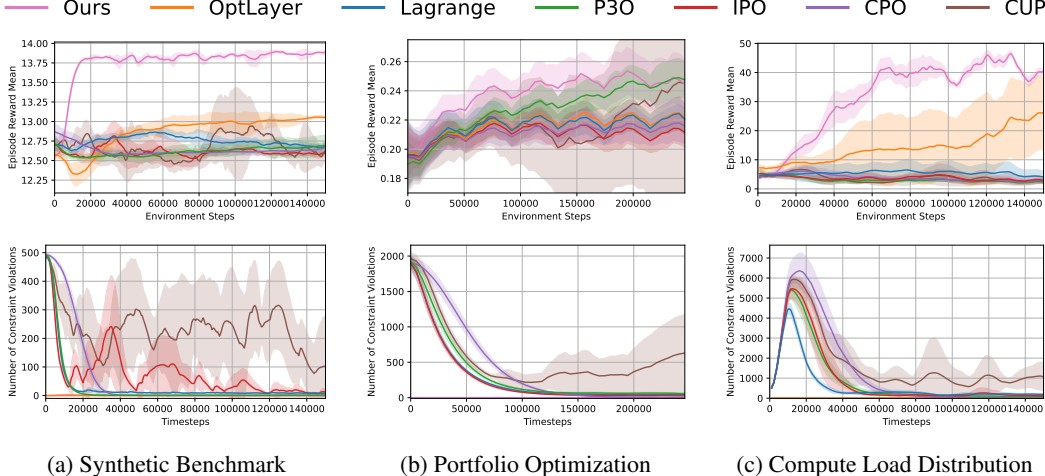

(a) Synthetic Benchmark     (b) Portfolio Optimization     (c) Compute Load Distribution

Figure 4: **Learning curves of all methods in three environments.** The x-axis corresponds to the number of environment steps. The y-axis is the average episode reward (first row), and the number of constraint violations during every epoch (second row). For portfolio optimization (b) we report the performance running eight evaluation on 200 fixed market trajectories. This is because in training, every trajectory is different which makes comparisons hard. Curves smoothed for visualization.

the number of affected entities for each constraint. We then sample the constraint coefficients from the range $[0, 1]$.

**Synthetic Environment** In addition to the aforementioned environments, we propose a synthetic benchmark. The reward surface consists of an MLP with random weights. Each episode compromises two states. As it is completely deterministic, it provides a simple yet effective way to benchmark approaches for constrained allocation tasks. In this setting, we create the constraints by randomly sampling 30 points and use their convex hull as the polytope defining the action space. We utilize a seven-dimensional setting with 611 constraints in our experiments.

## 5.1 Experimental Setup

We train PASPO using PPO [22] and compare our approach to various baselines, including state-of-the-art approaches for constrained allocation tasks and Safe RL. Specifically, we compare PASPO with five representative approaches from Safe RL: CPO [2], CUP [30], IPO [14], P3O [31], and PPO with Lagrangian relaxation. Additionally, we compare our method to OptLayer [18], a popular projection-based method for linear hard constraints. To maintain a consistent and fair comparison across different methods, we use the same hyperparameters across the different methods if possible. Many Safe RL approaches have difficulties handling equality constraints [11]. Therefore, we use a Dirichlet distribution to represent the policy in the baselines, thereby ensuring satisfaction of the simplex equality constraint. We do not share the parameters between the policy and value function. We use a fully-connected MLP with two hidden layers of 32 units and ReLU non-linearities for each policy, cost, and value function. In our approach, the state encoder and each policy head consists of a two-layer MLP. The training process is run for 150,000 steps and the results are averaged over five different seeds. In the portfolio optimization task, we use ten different seeds due to the stochasticity of the financial environment and train for 250,000 steps. Given the relatively small network sizes, training is conducted exclusively on CPUs. We implement our algorithm and the baselines using RLlib and PyTorch. More details regarding the environments, training, and hyperparameters can be found in the Appendix.

### 5.1.1 Performance of PASPO

We visualize the performance and constraint violations of all methods across our three environments in Figure 4. A tolerance of $1e^{-3}$ is used for evaluating constraint violations and we report the total number of violations per episode. In all three environments, PASPO converges faster to a higher

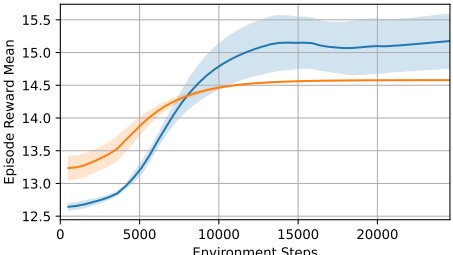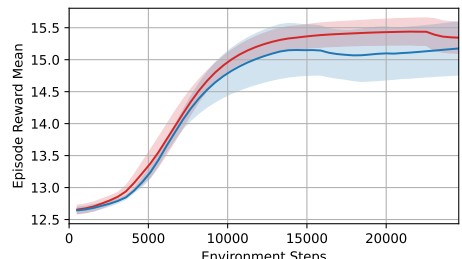

Figure 5: **Ablations** in (a) show the performance of our approach with (blue) and without (orange) the de-biased initialization. In (b) depicts the impact of the allocation order. We reverse the allocation order (red).

average return compared to baselines. Additionally, while all compared soft-constraint methods display constraint violations, only the hard constraint approaches PASPO and OptLayer guarantee to permanently satisfy the constraints. Finally, we can observe that the variance of PASPO is rather low compared to other methods. However, in portfolio optimization task (b) our approach displays some variance which we attribute to the stochasticity of the environment. Overall, these results demonstrate that our approach is not only able to consistently outperform other algorithms in terms of rewards but also guarantees no constraint violations.

### 5.1.2 Importance of de-biased Initialization and Order

We conduct ablation studies to investigate the impact of our de-biased initialization and the order of entity allocation on our synthetic benchmark. No constraints are applied except for the simplex constraint to highlight the effects. The results, shown in Figure 5, indicate that without de-biased initialization (orange in (a)), learning is slower and converges prematurely to a sub-optimal policy. In (b), we explore the impact of allocation order by reversing it (red) and observe no significant performance difference. This indicates that our approach is robust to the allocation order due to the use of the de-biasing initialization.

## 6 Limitations and Future Work

While PASPO guarantees that constraints are always satisfied, it is considerably more computationally expensive than standard neural networks in allocation tasks with many entities, as the sampling of each action requires solving a series of linear programs. RL in high-dimensional continuous action spaces is a very challenging task. Our approach cannot overcome this issue and also struggles in very high-dimensional settings. For future work, we plan to extend PASPO to also incorporate state-dependent constraints. While we evaluate our approach only on benchmarks with hard constraints, it can be applied to settings with both hard and soft cumulative constraints. In these scenarios, our method for handling hard constraints can be easily combined with most Safe RL algorithms to handle soft cumulative constraints.

## 7 Conclusion

In this paper, we examine allocation tasks where a certain amount of a resource has to be distributed over a set of entities at every step. This problem has many applications like logistics tasks, portfolio management, and computational workload processing in distributed environments. In all these applications, the set of feasible allocations might be bound by a set of linear constraints. Formally, these restrict the action space to a convex polytope. To define a stochastic policy function that can be used with policy gradient methods in RL, we propose an autoregressive process that computes allocation sequentially. We employ linear programming to compute the range of feasible allocations for an entity given the already fixed allocations of other entities. Our policy function consists of a sequence of one-dimensional beta distributions where the shape parameters $\alpha$ and $\beta$ are learned by neural networks. To counter the effect of initialization bias, we utilize a de-biasing mechanism to ensure sufficient exploration and prevent premature convergence to a sub-optimal policy. In our

experiments, we demonstrate that our novel method PASPO yields better results than state-of-the-art approaches while not having any constraint violations. Furthermore, we show that our initialization method yields better results than random initializations and counters the impact of the allocation order.

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

## A    Environments

The implementation for all three environments can be found at: `https://github.com/niklasdbs/paspo`.

### A.1    Financial Environment

The financial environment used for testing our approach is based on [28]. The financial market trajectories in this environment are sampled from a hidden Markov model, which was fitted based on real-world NASDAQ-100 data from January 3rd, 2011 to December 1st, 2021. The environment offers differently sized data sets of randomly selected assets contained in the NASDAQ-100. The experiments in the financial environment for this paper are run with 13 assets, which corresponds to

the *model_parameter_data_set_G_markov_states_2_12* data set with the *include_cash_asset=true* option. We initialize the environment with *seed=2*.

Table 1 shows a list of the assets used in the experiments.

For the experiments a random combination of two types of constraints typical for financial tasks is used: (a) a randomly selected subset of assets to either stay above or below an randomly selected allocation threshold and (b) thresholds for financial or environmental portfolio measures that can be calculated as an weighted average of the assets' individual measures, i.e., as a weighted linear combination. A list of these measure can be found in Table 2.

The experiments include 5 constraints, of which the number constraints of type (a) and type (b) is randomly decided. The exact implementation can be found in our code polytope_loader.py (generate_random_fin_env_polytope_rejection_sampling). We use the seed 2 to generate the constraints.

| Index | ISIN | Ticker | Name |
|-------|------|--------|------|
| 1 | - | CASH | CASH |
| 2 | US5949181045 | MSFT | Microsoft Corporation |
| 3 | US0567521085 | BIDU | Baidu Inc. |
| 4 | US00724F1012 | ADBE | Adobe Inc. |
| 5 | US6937181088 | PCAR | Paccar Inc. |
| 6 | US67066G1040 | NVDA | NVIDIA Corporation |
| 7 | US8552441094 | SBUX | Starbucks Corporation |
| 8 | US4612021034 | INTU | Intuit Inc. |
| 9 | US0530151036 | ADP | Automatic Data Processing Inc. |
| 10 | US0231351067 | AMZN | Amazon.com Inc. |
| 11 | US2786421030 | EBAY | eBay Inc. |
| 12 | US0311621009 | AMGN | Amgen Inc. |
| 13 | US7475251036 | QCOM | Qualcomm Inc. |

Table 1: List of assets used in the environment.

| | Est. Total Energy Use To EVIC USD in million | Est. Total $CO_2$ Equivalent Emissions To EVIC USD in million | Est. Weighted Average Cost of Capital, (%) | Est. Dividend yield, (%) | Est. Return On Equity, (%) |
|---|---|---|---|---|---|
| **CASH** | 0.00 | 0.00 | 0.00 | 0.00 | 0.00 |
| **MSFT** | 23.49 | 1.75 | 8.19 | 1.89 | 40.15 |
| **BIDU** | 70.80 | 15.17 | 7.12 | 0.00 | 9.01 |
| **ADBE** | 12.98 | 1.12 | 6.81 | 2.28 | 92.49 |
| **PCAR** | 83.66 | 10.27 | 7.31 | 3.14 | -43.29 |
| **NVDA** | 17.17 | 1.58 | 6.20 | 3.00 | 127.85 |
| **SBUX** | 85.73 | 6.79 | 7.24 | 1.13 | 26.61 |
| **INTU** | 41.23 | 17.15 | 7.87 | 0.00 | 21.50 |
| **ADP** | 1.70 | 0.15 | 8.12 | 0.56 | 22.46 |
| **AMZN** | 4.69 | 0.39 | 8.28 | 0.00 | 44.48 |
| **EBAY** | 3.49 | 0.30 | 9.35 | 0.02 | 80.24 |
| **AMGN** | 33.87 | 3.29 | 7.09 | 0.73 | 37.76 |
| **QCOM** | 47.07 | 4.09 | 8.33 | 2.16 | 36.06 |

Table 2: KPI estimates for assets based on 2021 (final year of the used data set, source: Refinitiv); EVIC - Enterprise value including Cash

## A.2 Compute Environment

The compute environment used for testing our approach is based on [3]. The agent's task is to allocate compute jobs to a given set of servers in a data center. A reward is triggered for each job that was completed in a predetermined maximum allowed computation time. The challenge of this environment is that the agent needs to match the queue of jobs still to be allocated with the different computational capabilities of the servers as well as each server's individual queue of jobs still to be computed. It is assumed that the compute jobs in the environment can be arbitrarily split and computed in parallel. The creation of new compute jobs is triggered by $n$ users and follows a Poisson process. A job is defined by its *payload size*, i.e., the data to be transferred to a server, its *required CPU cycles* for the processing workload, and its maximum allowed time until the job needs to be completely processed. These attributes for the jobs that can be created by each user are randomly sampled at creation of the environment.

The experiments in this paper run with a setup of 9 servers and 9 users that generate compute jobs. The parameter set used is *parameter_set_9_9_id_0*. We initialize the environment with *seed=1*. The randomly sampled specifications for the nine servers can be found in Table 3 and the job attributes created by the nine users can be found in Table 4.

To generate the constraints, we first sample the number of affected entities between 2 and 8 for each constraint and randomly choose the affected entities accordingly. We then uniformly sample constraint coefficients from the interval $[0, 1]$, as well as a corresponding constraint limit between 0 and 1. We use a seed of 1 to generate 5 constraints. The implementation can be found in polytope_loader.py (generate_random_polytope_rejection_sampling).

| Index | Max Compute Cycles per Second |
|-------|------------------------------:|
| 1 | 2 836 258 583 |
| 2 | 855 913 878 |
| 3 | 652 109 364 |
| 4 | 789 819 414 |
| 5 | 3 187 852 760 |
| 6 | 974 311 629 |
| 7 | 2 005 143 973 |
| 8 | 1 481 875 307 |
| 9 | 2 216 715 088 |

Table 3: Server Specifications

| User | Data Size in Bits per Job | Required Compute Cycles per Job | Average Number of Jobs Created per Interval | Interval Length in Seconds |
|------|---------------------------|---------------------------------|---------------------------------------------|----------------------------|
| 1 | 587 168 | 1 690 694 | 10 | 0.01 |
| 2 | 240 447 | 1 092 255 | 10 | 0.01 |
| 3 | 257 396 | 867 139 | 10 | 0.01 |
| 4 | 364 400 | 819 594 | 10 | 0.01 |
| 5 | 387 953 | 3 463 247 | 10 | 0.01 |
| 6 | 309 269 | 2 300 810 | 10 | 0.01 |
| 7 | 44 420 | 1 129 119 | 10 | 0.01 |
| 8 | 318 062 | 1 092 402 | 10 | 0.01 |
| 9 | 490 880 | 1 044 736 | 10 | 0.01 |

Table 4: User/Job Specifications

## A.3 Synthetic Benchmark

In addition to these environments, we propose a synthetic benchmark. Its reward surface consists of an MLP with random weights. An example of the reward surface in three dimensions is visualized in Figure 6 Each episode has two states. Since it is completely deterministic, it provides a simple but

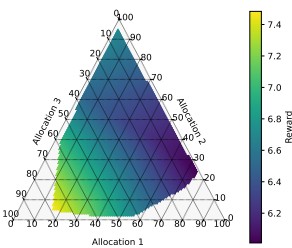

Figure 6: Example of the reward surface of our synthetic benchmark in three dimensions under constraints.

effective way to benchmark approaches for constrained allocation tasks. In this setting, we create the constraints by randomly sampling 30 points and use their convex hull as the polytope defining the action space. We use a seven-dimensional setting with 611 constraints in our experiments. More specifically, the network has one hidden layer and ReLU as a non-linearity. The input layer receives the state (as a number, i.e., 0 or 1) and the action as input and has an output size of 32, the hidden layer has an input size of 32 and output size of 16. The output layer has an input size 16 and a output size of 1. The exact initialization of the neural network weights can be found in our code (synth_env.py: MLPRewardNetwork). To generate the environment we use the seed 1 in our experiments.

To generate the constraints, we sample 30 randomly from a Dirichlet distribution with concentration parameters set to 1. We then build the convex hull of these points and convert the resulting polytope into its halfspace representation, i.e., a system of linear inequalities which we use as constraints. We use the seed of 1 to generate the constraints. This results in 611 constraints. The algorithm to generate the constraints can be found in the code (random_polytope_generator.py)

## B  Architecture

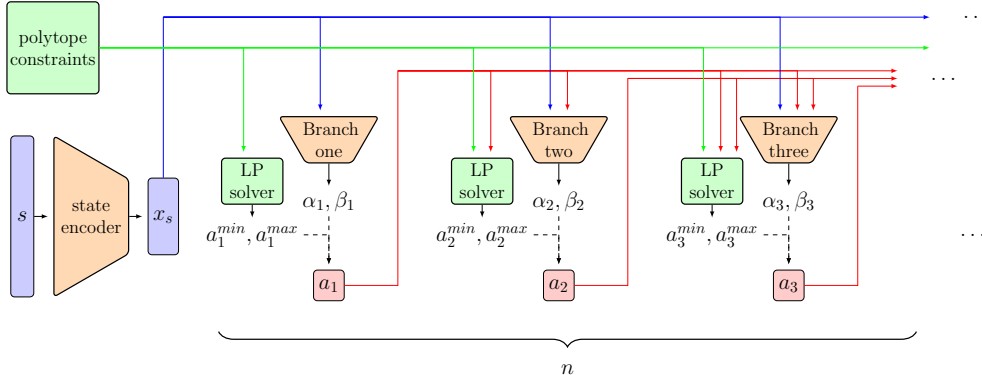

Figure 7: Architecture of PASPO

## C  Hyperparameters/Training

In Table 5 we list the most important parameters and hyperparameters. The full configurations used can be found in the config files (yaml/hydra based) in our code (run configs directory). We tuned hyperparameters on our synthetic benchmark with five dimensions and five constraints.

We do not train using GPUs because of the small network sizes. We used an internal CPU cluster with consumer machines and servers ranging from 8 to 90 cores and RAM between 32GB and 512GB.

| Parameter | Ours | IPO | P3O | CUP | Lag. | OptLayer | CPO |
|---|---|---|---|---|---|---|---|
| Training env steps | 150,000 (synth, compute), 250,000 (portfolio optimization) | | | | | | |
| Episode/Rollout length | 512 environment steps | | | | | | |
| Number of parallel envs | 8 | | | | | | |
| Learning Rate | 1e-3 | 1e-3 | 1e-3 | 1e-3 | 1e-3 | 1e-3 | – |
| Gradient clipping | 2.0 | | | | | | |
| Minibatch size | 64 | | | | | | |
| Optimizer | Adam | | | | | | |
| GAE lambda | 0.95 | | | | | | |
| Discount factor | 1.0 | | | | | | |
| No. grad update it per epoch | 10 (CPO only for the critic 40) | | | | | | |
| PPO clip parameter | 0.3 | 0.3 | 0.3 | 0.3 | 0.3 | 0.3 | – |
| Entropy coefficient | 0.01 | 0.01 | 0.01 | 0.01 | 0.01 | 0.01 | – |
| Cost limit | – | 1e-3 | 0.0 | 0.0 | 0.0 | 0.0 | 0.0 |

Table 5: The most important Parameters and Hyperparameters for Various Methods

## D  Guaranteed Constraint Satisfaction

In the following, we proof that our approach PASPO always guarantees constraint satisfaction and that our method is able to sample all possible actions from the constrained action space.

**Definitions:** Let $A$ be the set of all actions that can be sampled with PASPO, and let $P = \{a \in \mathbb{R}^n | Ca \leq b\}$ be the convex polytope that corresponds to constrained action space. We define $A_*^{(i+1)} = \{a \in \mathbb{R}^n | C^{(i+1)}a^{(i+1)} \leq b_*^{(i+1)}, \forall j = 1, \ldots, i : a_j = a_j^*\}$ where $b_*^{(i+1)} = b -$

$\sum_{j=1}^{i} a_j^* \begin{bmatrix} c_{1j} \\ \vdots \\ c_{mj} \end{bmatrix}$ and $C^{(i+1)}$ and $a^{(i+1)}$ as defined in the paper.

Thus, $A_*^{(i+1)}$ is the restricted action space after sampling/fixing already the allocations $a_1^*, \ldots, a_i^*$.

**Theorem 1.** *Let $P = \{a \in \mathbb{R}^n | Ca \leq b\} \neq \emptyset$ be the convex polytope that corresponds to a constrained action space. Let $A$ be the set of all the points that can be generated by PASPO. It holds that $A = P$.*

*Proof.* Well-defined: Show that $A^{(n)} \neq \emptyset$ if $P \neq \emptyset$.

Induction over $i$:

$i = 1:$     $A_*^{(1)} = \{a \in \mathbb{R}^n | C^{(1)}a^{(1)} \leq b_*^{(i+1)}\} = \{a \in \mathbb{R}^n | Ca \leq b\} = P \neq \emptyset$

$i \to i+1:$

$(i+1 \leq n)$  $A_*^{(i)} \neq \emptyset \Rightarrow \exists a^\uparrow, a^\downarrow \in A_*^{(i)} : a_i^\uparrow = a_i^{\min}, a_i^\downarrow = a_i^{\max}$

Now assume an arbitrary $a_i^*$ is sampled from $[a_i^{\min}, a_i^{\max}]$

$\Rightarrow \exists \lambda \in [0,1] : a_i^* = \underbrace{(\lambda a^\downarrow + (1-\lambda)a^\uparrow)}_{:=a^\lambda}{}_i$

By convexity of polytopes as solution spaces for linear inequality systems, we get:

$$\begin{bmatrix} c_{1,i} & \cdots & c_{1,n} \\ \vdots & \ddots & \vdots \\ c_{m,i} & \cdots & c_{m,n} \end{bmatrix} \begin{bmatrix} a_i^\lambda \\ \vdots \\ a_n^\lambda \end{bmatrix} \leq b - \sum_{j=1}^{i-1} a_j^* \begin{bmatrix} c_{1j} \\ \vdots \\ c_{mj} \end{bmatrix} \underset{(a_i^\lambda = a_i^*)}{\Longleftrightarrow}$$

$$\begin{bmatrix} c_{1,i+1} & \cdots & c_{1,n} \\ \vdots & \ddots & \vdots \\ c_{m,i+1} & \cdots & c_{m,n} \end{bmatrix} \begin{bmatrix} a_{i+1}^\lambda \\ \vdots \\ a_n^\lambda \end{bmatrix} \leq b - \sum_{j=1}^{i} a_j^* \begin{bmatrix} c_{1j} \\ \vdots \\ c_{mj} \end{bmatrix} \Rightarrow a^\lambda \in A_*^{(i+1)}$$

To show that $A = P$:

$A \subseteq P$ :          Let $a^* \in A$. In the last step $(n)$, $a_n^*$ is sampled (by design) such that

$$C^{(n)} a_n^* \leq b - \sum_{j=1}^{n-1} a_j^* \begin{bmatrix} c_{1j} \\ \vdots \\ c_{mj} \end{bmatrix} \Leftrightarrow C a^* \leq b \Leftrightarrow a^* \in P$$

$A \supseteq P$ :          Let $a^* \in P. \Leftrightarrow C a^* \leq b \Leftrightarrow C^{(i)} a^* \leq b^{(i)} \ \forall i \Leftrightarrow a^* \in A_*^{(i)} \ \forall i$
                         $\Rightarrow$ We can construct $a^*$ by sampling $a_i^*$ in every step $i$. $\Rightarrow a^* \in A$

$\square$

The intuition of why our approach can guarantee the satisfaction of constraints is based on three properties that we utilize: (1) If $P_i$ is the set of solutions to an system of linear inequalities, then by adding further constraints to the system of linear inequalities there will be a new set of solutions $P_{i+1}$ but always such that $P_{i+1} \subseteq P_i$. (2) For any two points $a^{\min}$ and $a^{\max}$ in a convex set it can be implied that there exists a point $a^\lambda$ for which the following is true for its i-th dimension $\exists \lambda \in [0,1] : a_i^* = \underbrace{(\lambda a^{\min} + (1 - \lambda) a^{\max})}_{:=a^\lambda}{}_i$ (3) Linear Programming can determine the upper and lower bounds for single variables in a system of linear inequalities, i.e. $[a_i^{\min}, a_i^{\max}] \forall i$.

We start with the original system of linear inequalities with the solution space $P_1 \neq \emptyset$ which is a convex polytope. We use (3) on $P_1$ to determine the upper and lower bounds for $a_1$, i.e. $[a_1^{\min}, a_1^{\max}]$. We sample a value $a_1^*$ from the range $[a_1^{\min}, a_1^{\max}]$. We know that the solution space $P_1$ must contain at least one point for which in its 1st dimension $a_1 = a_1^*$ due to (2). In the next step we add the further constraint $a_1 = a_1^*$ to the system of linear inequalities. This updated system of linear inequalities will have the solution space $P_2$. Due to (2) $P_2 \neq \emptyset$, as well as $P_2 \subseteq P_1$. We then repeat the entire process and use (3) on $P_2$ to determine the upper and lower bounds for $a_2$, i.e. $[a_2^{\min}, a_2^{\max}]$....

After the n-th iteration $a_n^*$ will be determined and we then have completed the generation of point $a^* = (a_1^*, ..., a_n^*) \in P_n \subseteq ... \subseteq P_1$, i.e. we succeed generating a point $a^*$ that satisfies all original constraints $P_1$.

# E   The Impact of the Allocation Order

As already discussed in the ablations in the main paper, with our de-biased initialization the impact of the allocation order is small. However, without our de-biased initialization, the order of the allocation has a significant impact on the performance, as illustrated in Figure 8.

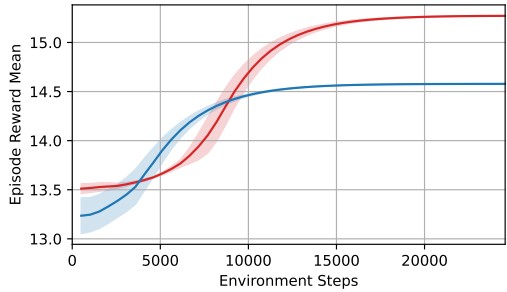

Figure 8: The impact of the allocation order on PASPO without de-biased initialization in the synthetic benchmark with two states, a 7-dimensional action space, and no additional allocation constraints. Blue depicts the standard allocation order (i.e., $e_1, e_2, \ldots, e_n$) and red depicts the reversed allocation order (i.e., the entities are allocated in the reversed order). A significant difference in performance can be observed with respect to the order without our de-biased initialization. In contrast, Figure 5b in the paper shows that with the de-biased initialization the difference is not significant.

