# OpenReview forum: "Autoregressive Policy Optimization for Constrained Allocation Tasks"
_NeurIPS.cc/2024/Conference — NeurIPS 2024 poster_

### Official Review · Reviewer_2peK · 2024-07-10

**Soundness:** 3
**Presentation:** 3
**Contribution:** 3
**Rating:** 5
**Confidence:** 2

**Summary:**

This paper studies task allocation under resource constraints and proposes a new constrained RL algorithm based on autoregressive policy optimization with a novel de-biasing mechanism. Extensive simulations are provided demonstrating the improved performance.

**Strengths:**

The paper is well-written and the algorithm design is novel.

The numerical performance of the proposed algorithm is much better than the benchmark: no constraint violation but enjoys better reward than benchmark algorithms that allow constraint violation.

**Weaknesses:**

1. Is there any theoretical guarantee to explain no constraint violation in the simulation? It is quite surprising to me that the proposed algorithm has no constraint violation since the problem formulation allows constraint violation. Can the authors provide some intuitions on such a good constraint satisfaction behavior in the simulation results?

2. In Section 4.1, does the order of the constraints affect the performance (for example, if we first determine the feasible interval for constraint a_2, then determine a_1, will it cause a difference in the performance)? If there is no difference, what is the reason? If there are differences in performances, how to choose an (near) optimal ordering of the constraints to improve the performance?

**Questions:**

See above.

**Limitations:**

The authors discussed the limitations explicitly in the second last section.

---

> ### Author Rebuttal · Authors · 2024-08-06
>
> Thank you for the positive review and helpful comments. We are happy to address your questions below:
>
>
> - W1: Please see our general response, where we address the theoretical guarantees in the form of a proof.
> - W2: In theory, it is possible to learn the optimal policy regardless of the order as long as the policy is able to sample the complete action space. In practice, however, this might not be the case when the initialization of the policy does not uniformly sample over the action space, it might not be able to overcome this bias and converge early to a suboptimal policy because large allocations for later entities are not sufficiently explored. Instead of looking for the optimal order, we provide a de-biasing mechanism that mostly eliminates the impact of the order, which can be seen in our ablations (Figure 5b). There, we show that the order does not significantly impact the performance of our approach when using our de-biasing mechanism. We performed an additional experiment to also compare the impact of the order without using our de-biasing mechanism. This figure (which can be found in the PDF attached to our general response) shows that reversing the order without our de-biasing mechanisms does create a significant difference in performance.

---

> > ### Comment · Reviewer_2peK · 2024-08-08
> >
> > Thanks for your responses and the theoretical discussions! However, I think it is better to resubmit this paper next time with the theoretical results added. So I will keep my score.

---

### Official Review · Reviewer_3WTq · 2024-07-10

**Soundness:** 3
**Presentation:** 3
**Contribution:** 2
**Rating:** 5
**Confidence:** 2

**Summary:**

he paper presents Polytope Action Space Policy Optimization (PASPO), a novel RL methodology tailored for strict linear constraint allocation tasks. By autoregressively decomposing the action space into manageable sub-problems, PASPO enhances compliance and efficiency without the need for corrective actions. Additionally, an innovative de-biasing mechanism during early training prevents premature policy convergence, showing significant improvements over existing methods in constraint adherence and task performance.

**Strengths:**

The paper is very well-written.
The idea of PASPO is novel and interesting.
The motivation of the problem is clear and have practical significance.
The experiment results are clear and performs well.

**Weaknesses:**

1. The algorithm requires substantial computing resources, demanding high-performance computational capabilities to effectively manage its intensive processing needs. This entails advanced processors, ample memory, and significant data storage capacity to efficiently handle and execute the complex calculations involved. I appreciate that the authors also highlighted this in the article.

2. The paper lacks a comprehensive explanation of why the algorithm achieves high performance, omitting essential details on its operational principles and optimization techniques. Additionally, there is insufficient exploration of the algorithm's adaptive features, error handling, and scalability under varying conditions. These aspects are critical for understanding its potential effectiveness across diverse scenarios and applications.

3. Given that the allocation problem is formulated as a clear CMDP problem, I am curious whether the authors could provide theoretical guarantees for their approach. As I am not well-versed in this topic, I will consider the insights of other reviewers on this matter.

**Questions:**

none

---

> ### Author Rebuttal · Authors · 2024-08-06
>
> Thank you for the positive review and helpful comments. We are happy to address your questions below:
>
> - W1: Thank you for appreciating that we discussed this in our paper. Regarding the computational requirements, see also our response to reviewer QtCd question 1.
> - W2: While several reviewers like the presentation and clarity of our paper, they noted that minor details were missing or unclear, which we address in our responses. We also added a proof to show that our approach guarantees constraint satisfaction (see our general response). Furthermore, our formulation of the problem as a CMDP with cost functions caused some confusion, since our approach does not require cost functions. We did this to show the relation to Safe RL and will clarify this in our paper (see also our response to reviewer 3hEE question 4). In case our response did not cover your concerns, could you please point us to more specific questions, we are happy to hear and address them in the discussion.
> - W3: Please see our general response, where we address the theoretical guarantees in the form of a proof.

---

> > ### Comment · Reviewer_3WTq · 2024-08-13
> >
> > Thanks for the responses. I do not have further questions.

---

### Official Review · Reviewer_QtCd · 2024-07-13

**Soundness:** 3
**Presentation:** 4
**Contribution:** 3
**Rating:** 7
**Confidence:** 1

**Summary:**

This work considers constrained allocation tasks, like portfolio optimization or server scheduling and focuses on how to do policy optimization while respecting the given constraints.

In these problems, the core problem is that sampling over the polytope is challenging.

The work puts forward an approach for sampling points that satisfy the constraints which can be used in known policy iteration algorithms.

The approach is logical: iterate over dimensions, determining feasible range and then random sampling. Careful choice and learning of distribution parameters for sampling leads to good outcomes, removing the bias resulting from fixing earlier parameters first. This is used as the core sampling method within Proximal Policy Optimization training.

**Strengths:**

The core sampling approach PASPO is presented in a very clear way, and the sample approach is sensible and quite simple in a complicated space.

The authors compare to prior approaches are are able to show that the PASPO approach is more effective and trains faster than prior work.

The approach seems to improve significantly on other RL based approaches to given problems, though I am not close enough to the area to validate that.

**Weaknesses:**

The sampling approach put forward is clear and straightforward. At the same time, it is not clear the level of full novelty involved in the solution - it is a greedy sampling approach with carefully constructed weighting to avoid bias. With that said, if that gets the improved results, then much better to have that come from a simple solution than a much more complicated solution.

**Questions:**

You discuss the high computational cost of your approach - can you discuss it in the context of the other RL approaches compared against in Figure 4?

**Limitations:**

The authors have adequately addressed limitations of the work.

---

> ### Author Rebuttal · Authors · 2024-08-06
>
> Thank you for the positive review and helpful comments. We are happy to address your questions below:
>
>
> - W1: The novelty of our approach lies in utilizing the properties of the polytope to efficiently parameterize a stochastic policy over it in an autoregressive way that can be optimized using standard reinforcement learning algorithms like PPO. Additionally, we introduce a novel de-biasing mechanism. We are happy that the reviewer acknowledges the good results that our approach achieves.
> - Q1: As discussed in the paper, our approach introduces some additional computational cost, since we are required to solve linear programs in our approach. Our approach does not have particularly demanding hardware requirements and can be trained on regular servers, PCs, or laptops. It is difficult to directly compare the runtimes because we trained on a cluster with various different servers and also ran multiple training runs in parallel, which impacted the duration of training. Therefore, we were hesitant to report exact numbers in the paper. Note that the runtime also depends on the number of constraints and dimensions of the action space and we did not optimize our implementation for runtime. For example, in our synthetic benchmark of dimensionality 7 and 150k environment steps, OptLayer took around 24h to train and our approach also took around 24h. In contrast to the other benchmark approaches, OptLayer can also guarantee constraint satisfaction. Approaches like IPO, CPO, etc, take around 10 minutes but do not guarantee constraint satisfaction at all times. Let us also note that we trained only on CPUs and used scipy to solve the linear programs. Note that works exist that utilize GPUs for solving linear programs (for example see [1]), which could be utilized to drastically speed up training.
>
> [1] Spampinato, Daniele G., and Anne C. Elster. "Linear optimization on modern GPUs." 2009 IEEE International Symposium on Parallel & Distributed Processing. IEEE, 2009.

---

### Official Review · Reviewer_3hEE · 2024-07-13

**Soundness:** 3
**Presentation:** 3
**Contribution:** 3
**Rating:** 5
**Confidence:** 3

**Summary:**

The paper presents a method for a specific setting of constrained RL method. The setting deals with constraints on the simplex of action space. The authors motivate this setting with resource allocation problems. The proposed method uses sequential conditional sampling of actions to impose constraints and also they propose a de-biasing mechanism to be robust against initialization.

**Strengths:**

- The paper is clearly written and easy to understand with references made to figures often
- Presents an algorithm for Hard constraint type of problems and achieves good empirical performance.
- de-biasing mechanism

**Weaknesses:**

- The paper is motivated by a requirement of guaranteed constrained satisfaction however the result is empirical and not theoretical guarantees of constraint satisfaction.

- The algorithm is explained in words in different parts of the paper but it misses an overall algorithm or flow of the framework.

- If I had to solve the same problem (Maximizing an objective while having joint constraints on the action space), I would directly sample actions from the constrained simplex and apply an RL algorithm, e.g., PPO to it (i think you can do rejection sampling to sample from constrained simplex). Am I missing something? How would this perform as compared to PASPO? Also, it would be nice if you could explain in detail why you think one would work better than the other.

**Questions:**

- Since the constraint satisfaction is empirical, I would ask authors to not use the word guarantee for it.
- While reading the paper, I had a question of how much de-biasing helps. It's nice that the authors did an ablation study.  Which environment is used for the ablation study? And do you expect similar performance for all the environments?
- In Figure 5 b) what does reverse allocation order mean?
- What is the final algorithm? What is the objective of PPO and do you impose any constraints as you describe in lines 135 and 133? or are constraints indirectly applied by restricting the action sampling?
- Why do you use the word auto-regressive?
- why do you sample actions sequentially and not jointly all at once?
- What are the state and action space dimensions of the environments used for empirical study?

- comment to improve the paper (optional)
I understand that resource allocation is an application where the framework directly fits. However, satisfying hard/state/action-wise constraints is also not so trivial with general constraint RL algorithms. A possibility could be that the authors do not directly focus on allocation tasks but rather on a framework for safe RL with a particular type of constraints and later motivate is with application of resource allocations.

---

> ### Author Rebuttal · Authors · 2024-08-06
>
> Thank you for the positive review and helpful comments. We are happy to address your questions. We try to be as detailed as possible given the 6k limit.
>
>
> - W1/Q1: Please see our general response regarding the theoretical guarantees.
> - W2: Thank you for the suggestion. To assist readers in understanding the overall flow of our approach, we will include a detailed algorithm and a visualization of the flow.
> - W3: Sampling from the constrained simplex is not sufficient for RL. We need to define a stochastic policy over the polytope and update its parameters using a gradient in order to increase or decrease the probability of drawing these samples. Parameterizing such a differentiable distribution via rejection sampling is not straightforward. However, some algorithms use a related concept. They sample from a distribution over a superset, e.g., an unconstrained polytope, but instead of rejecting actions that fall outside the constrained polytope they project these onto the constrained polytope. OptLayer [18] is a prominent instance. However, action projections can lead to a biased policy gradient [15]. In contrast, our approach defines a stochastic policy only over the polytope and does not suffer from this problem. The experimental results also empirically demonstrate that we can outperform projection-based approaches like OptLayer.
> - Q2: We use the synthetic benchmark without additional allocation constraints (i.e., only the simplex constraint) for our ablation. As in our main experiments, this environment is 7-dimensional. Since this was not sufficiently clear in the paper, we will add more details regarding the setting to the paper. We decided not to use additional allocation constraints because this makes the effect of de-biased initialization most pronounced since each constraint limits the possible allocations (e.g., if one is required to allocate at least 90% to one entity, there is not a lot of choice for the remaining allocations). Since the bias can occur regardless of the environment, we expect a similar performance in other environments.
> - Q3: We will clarify this in the paper. To derive a considerably different allocation order, we reverse the allocation order. That means, instead of allocating to entity $e_1, \ldots, e_n$, we allocate in the order $e_n, \ldots, e_1$, i.e., we allocate the last entity first, the second last ($e_{n-1}$) second, and so on.
> - Q4:
>  > “do you impose any constraints as you describe in lines 135 and 133? [...]”
>
>   We apologize for the confusion. The last paragraph of section 3 is somewhat misplaced as it describes how the problem setting can be reformulated to apply methods from Safe RL. Thus, we will move this definition to the experiments. We do not use a cost function, instead we apply the constraints to the action space and our approach ensures that actions always comply with the constraints.
>   > “objective of PPO?”
>
>   Our approach defines a differentiable stochastic policy function over the constrained action space and can be directly optimized using the standard objective of PPO.
> - Q5: In our approach, each step depends on the outputs of all previous steps (cf. Figure 2 in the Appendix), similar to an autoregressive model. Hence, we use the word “autoregressive”.
> - Q6: RL algorithms that are policy gradient based, require sampling actions from the current policy, i.e., a distribution which is parameterized by a neural network. However, directly defining such a distribution over a polytope is extremely difficult. In fact, even efficiently generating uniform samples from a polytope is a difficult problem subject to ongoing research [8]. Therefore, our approach decomposes the problem into dependent sub-problems. Because of this autoregressive dependency, we need to sample an action sequentially. Note that this decomposition and process allows for training using standard RL algorithms.
> - Q7: We will add further details to the Appendix. In portfolio optimization the state space has 27 dimensions and the action space 13. In the computational workload distribution environment the state space has 18 dimensions and the action space 9.  In our synthetic env, we have a discrete number of states, which are one-hot encoded. Our synthetic env uses a complex reward surface which is well suited to study the properties of various approaches, especially how well and fast approaches can optimize an arbitrary reward surface. Therefore, we do not want to focus too much on other properties of RL that are mostly orthogonal to our approach, such as delayed rewards, learning complex state representations, etc. As a result, we set the number of states to two and use a 7-dimensional action space.
> - Q8: Our approach can be applied to all constrained RL tasks, where the action space is a convex polytope, as long as the constraints are explicitly given. However, since many people think of Safe RL in settings where the constraints are not explicitly given and typically only soft satisfaction is required, we decided to focus on allocation tasks. However, as shown in our problem definition, constrained allocation tasks can be defined within the framework of Safe RL and we discuss the relation to Safe RL in our related work.

---

### Author Rebuttal · Authors · 2024-08-06

We thank the reviewers for their detailed and constructive feedback. We will integrate many suggestions in the paper and are happy that our approach has been well received. Specifically, we are glad that reviewers find that:
- our paper is well-written and clearly presented:
  - “PASPO is presented in a very clear way” (QtCd)
  - “The paper is clearly written and easy to understand with references made to figures often” (3hEE)
  - “The paper is very well-written.” (3WTq)
  - “The paper is well-written” (2peK)

- our solution is effective and the experiments are well-conducted:
  - “The experiment results are clear and performs well.” (3WTq)
  - “Presents an algorithm for Hard constraint type of problems and achieves good empirical performance.” (3hEE)
  - “The authors compare to prior approaches are are able to show that the PASPO approach is more effective and trains faster than prior work.” (QtCd)
  - “The numerical performance of the proposed algorithm is much better than the benchmark: no constraint violation but enjoys better reward than benchmark algorithms that allow constraint violation.” (2peK)

- our paper presents a novel and significant contribution:
  - “The idea of PASPO is novel and interesting. The motivation of the problem is clear and have practical significance.” (3WTq)
  - “[...] the algorithm design is novel.” (2peK)


Before we address the questions raised by the individual reviewers in our specific responses to the individual reviewers, we would like to address a point that has come up in multiple reviews:

## Can we provide theoretical guarantees that our approach always ensures constraint satisfaction?
**We will add a proof to the paper that shows that our approach always guarantees constraint satisfaction and that our approach is able to generate every possible constraint-compliant action. The proof is included in the attached PDF-file to this response.**

In short, we show that the set of all actions $A$ that can be generated by PASPO is equivalent to the original constrained solution space $P$ of the system of inequalities.
We prove via induction that PASPO will always generate a point $a^*$ that satisfies all constraints if $P\neq \emptyset$. The intuition is that when we determine the allocation to a single entity (that means we fix the value of a single variable in the inequality system), we express this by adding further constraints to the original system of inequalities. Therefore, every solution for the resulting system of inequalities must also be a solution for the original system of inequalities.
Furthermore, we show that any point within $P$ can also be generated via PASPO and that $A=P$.

A geometric illustration of this process can be found in our paper in Figure 2.

In the individual responses, we refer to questions and weaknesses in order of their occurrence (i.e., W1 refers to the first weakness, Q1 refers to the first question).
Apart from the proof, the PDF attached below also contains the result of an additional experiment conducted to better answer the question of reviewer 2peK regarding the impact of the allocation order in our approach.

---

### Decision · Program_Chairs · 2024-09-25

**Decision:**

Accept (poster)

**Comment:**

This paper introduces a new technique for policy optimization in the context of constrained allocation tasks. The paper has several strengths. First, the problem is well motivated and of practical importance. Second, the approach taken is novel and creative. Third, the experimental analysis demonstrates the approach's effectiveness compared to previous approaches to this problem rather convincingly. Finally,  the paper is well written and easy to follow.

The paper also has some weaknesses. First, although it claims to produce solutions where no hard constraints are violated and the empirical analysis demonstrates this fact, no theoretical guarantee of this property is given in the paper. A proof of this property of the algorithm was provided during the rebuttal period, however, showing that for each sequential decision, the space that can be reached by PASPO is  the constrained sample space created by the previous decision. Other weaknesses identified include: (1 ) the presentation is unclear in specifying several important aspects of the PASPO algorithm, (2) the paper lacks a comprehensive explanation of why the algorithm achieves high performance, and omits essential details on its operational principles and optimization techniques, and (3) the approach has significant computational requirements in relation to prior approaches to this problem. Subsequent discussion during the rebuttal period argued that there would not be a significant runtime problem, and also proposed revisions to address other weaknesses. Please incorporate all of these revisions and the proof in your final version should this paper be accepted.